# A Ten-Year Retrospective Review of Medical Records of Patients Admitted with Meningitis or Encephalitis at Five Hospitals in the United States Highlights the Potential for Under-Ascertainment of Invasive Meningococcal Disease

**DOI:** 10.3390/pathogens14100962

**Published:** 2025-09-24

**Authors:** Julio Ramirez, Stephen Furmanek, Thomas Chandler, Josue Prado, Lisa R. Harper, Steven Shen, Raffaella Iantomasi, Jessica V. Presa, Mohammad Ali, Jamie Findlow, Jennifer C. Moïsi, Frederick J. Angulo

**Affiliations:** 1Norton Infectious Diseases Institute, Norton Healthcare, Louisville, KY 40203, USA; julio.ramirez@nortonhealthcare.org (J.R.); stephen.furmanek@nortonhealthcare.org (S.F.); thomas.chandler@nortonhealthcare.org (T.C.); josue.prado@nortonhealthcare.org (J.P.); 2Global Vaccines Medical Affairs, Pfizer, Collegeville, PA 19426, USA; lisa.r.harper@pfizer.com (L.R.H.); steven.shen@pfizer.com (S.S.); raffaella.iantomasi@pfizer.com (R.I.); jessica.presa@pfizer.com (J.V.P.); jamie.findlow@pfizer.com (J.F.); jennifer.moisi@pfizer.com (J.C.M.); 3Evidence Generation and Medical Affairs, Pfizer, Collegeville, PA 19426, USA; mohammad.ali2@pfizer.com; 4Medical Evidence Group, Pfizer, Collegeville, PA 19426, USA

**Keywords:** meningitis, specimen collection, laboratory diagnostics, cerebrospinal fluid, bacterial culture

## Abstract

Laboratory confirmation of invasive meningococcal disease (IMD) relies on detection of *Neisseria meningitidis* in a biological specimen. Clinical management guidelines for patients presenting with signs and/or symptoms of meningitis and encephalitis emphasize the need for appropriate specimen collection for laboratory testing. To explore the potential for IMD under-diagnosis, we reviewed medical records of patients admitted with signs and/or symptoms of meningitis or encephalitis at five hospitals in Louisville, Kentucky, in 2014 to 2023. Among 675 patients admitted with meningitis and/or encephalitis with cerebrospinal fluid (CSF) cultures who received antibiotics, 300 (44.4%) received antibiotics before CSF collection. Among 431 with blood cultures who received antibiotics, 133 (30.9%) received antibiotics before blood collection. Among 751 patients with CSF collected, 651 (86.7%) CSF specimens were tested using polymerase chain reaction (PCR) for *N. meningitidis* detection. No blood specimens were PCR-tested. These findings indicated that current standard-of-care practices may lead to IMD under-diagnosis. Since public health surveillance relies on IMD laboratory diagnosis, these findings highlight the potential for under-ascertained IMD by surveillance.

## 1. Introduction

IMD is a rare but serious illness caused by *Neisseria meningitidis* [1]. IMD can progress within hours from nonspecific symptoms, such as fever and nausea, to life-threatening manifestations that most commonly include meningitis and/or septicemia; other presentations of IMD include encephalitis and pneumonia [1,2,3,4]. Approximately 8% of IMD cases are fatal [5]. Of note, while many IMD patients present with meningitis, an important proportion of IMD patients present with septicemia without clinical signs or symptoms of meningitis or encephalitis. Additionally, a substantial percentage of IMD survivors experience permanent, disabling sequelae such as limb amputation, hearing loss, and learning disabilities [6].

Laboratory confirmation of IMD relies on detection of *N. meningitidis* in a specimen collected from a normally sterile site [1,3,7]. Bacterial culture of *N. meningitidis* from a specimen collected from a normally sterile site has historically been the gold standard for laboratory confirmation of IMD [1,3,8]. Accordingly, clinical management guidelines for patients with suspected meningitis recommend obtaining and performing diagnostic testing on both CSF and blood [9,10,11,12]. Culture has virtually 100% specificity for detecting *N. meningitidis* but can have low sensitivity [13,14], in part due to low concentrations of viable bacteria in blood and CSF [8]; moreover, culture sensitivity rapidly decreases when antibiotics are administered prior to specimen collection [13,15,16,17,18], as is often recommended in suspected IMD cases [1,19].

An alternative method for *N. meningitidis* detection in CSF and blood is PCR, which is more sensitive and less affected by antibiotic administration compared with culture [1,13,16]. Several studies have demonstrated that the frequency of detecting IMD increased using PCR testing compared to laboratory testing without PCR. For example, the incidence of laboratory-confirmed IMD (per 100,000 population per year) in Poland increased from 0.09 in 2002 to 0.76 in 2011 due, in part, to increased use of PCR testing; by 2011, 27.6% of 293 IMD cases were identified by PCR alone [20]. Furthermore, analyses of pediatric IMD cases admitted to hospitals in Spain in the early 2000s showed that 49.3% of the 75 IMD cases in one study and 39.3% of the 117 IMD cases in another study were identified by PCR alone [18,21].

Recently published World Health Organization (WHO) guidelines for management of patients with meningitis encourage CSF collection via lumbar puncture as soon as possible, preferably before commencing antibiotic administration, in the absence of contraindications or reasons for deferral; the guidelines also state that CSF testing should include bacterial culture and PCR-based molecular tests for relevant pathogens [10]. The WHO guidelines recommend obtaining blood samples for cultures as soon as possible, preferably before commencing antibiotic administration [10]. However, the WHO guidelines do not recommend routine use of blood PCR tests, indicating that blood PCR tests were not reviewed as part of the guideline assessment and stating that such tests can be used to enhance etiological confirmation when resources allow [10]. In the United Kingdom, more detailed IMD specific guidelines are available which align with the WHO meningitis guidelines but also incorporate routine recommendations for blood PCR testing in meningitis and septicemia cases [12]. By contrast, in the United States, PCR tests for detecting *N. meningitidis* in blood specimens are neither readily available nor recommended for patients with suspected IMD [22].

In the United States and many other countries, laboratory-confirmed IMD is a notifiable condition that is subject to national surveillance [22]. The comprehensiveness of IMD surveillance data depends on standard-of-care practices including appropriate specimen collection and laboratory testing. Failure to collect appropriate specimens in a timely manner (i.e., before antibiotic administration) from patients, or failure to appropriately test collected specimens, can lead to under-diagnosis of IMD [23,24,25,26], which in turn would lead to under-ascertainment by public health surveillance and ultimately underestimation of the disease burden. This study aimed to analyze IMD laboratory confirmation methods for patients presenting with signs and/or symptoms of meningitis and/or encephalitis and admitted to one of five hospitals in Louisville, Kentucky, and assess the potential for IMD under-diagnosis and potential implications for meningococcal disease surveillance.

## 2. Materials and Methods

### 2.1. Setting and Patients

This study was a retrospective review of the electronic medical records (EMRs) of patients with signs and/or symptoms of meningitis and/or encephalitis who were admitted to five Norton hospitals in Louisville, Kentucky from 2014 to 2023. The five participating hospitals have a combined total of 2204 beds and had almost 100,000 hospital admissions per year during the study period. The hospitals included Norton Audubon Hospital (432 beds), Norton Brownsboro Hospital (432), Norton Children’s Hospital (372), Norton Hospital (605), and Norton Women’s and Children’s Hospital (363).

Patients were included if they were admitted to the hospital with one or more of the admission codes, using the International Classification of Diseases, Tenth Revision (ICD-10) Diagnosis-Related Group (DRG) codes, for meningitis or encephalitis (Table 1). Corresponding clinical signs and symptoms (i.e., fever, headache, stiff neck, sensitivity to light, nausea, vomiting, confusion, etc.) recorded in the EMRs of these patients were reviewed by study physicians; patients without such signs and/or symptoms were excluded from the analysis. The following data were collected for all included patients: age; sex; underlying medical condition(s); CSF and blood collection and, if applicable, timing thereof; and receipt of intravenous (IV) antibiotics and, if applicable, timing thereof.

### 2.2. Laboratory Methods

During the study period, all five hospitals used the Norton central diagnostic laboratory. CSF specimens were collected into multiple tubes and immediately transported to the central diagnostic laboratory. CSF PCR was not available at the central diagnostic laboratory between 2014 and 2016; it could only be performed upon special request by sending the specimen to a reference laboratory (the specific PCR test used in 2014–2016 was not recorded). Since 2017, CSF samples from patients with suspected infection were tested at the Norton central diagnostic laboratory by PCR using the BioFire meningitis/encephalitis panel (bioMérieux, Marcy-l’Étoile, France), which includes *N. meningitidis*. Blood specimens were immediately inoculated into BacTecTM blood culture bottles (BD, Franklin Lakes, NJ, USA) and then transported to the central diagnostic laboratory. PCR testing of blood specimens for *N. meningitidis* was not performed at the Norton central diagnostic laboratory during the study period as no commercial test was available.

## 3. Results

A total of 1024 patients with hospital admission codes relating to meningitis and/or encephalitis were identified during the 10-year study period, of whom 988 (96.5%) had signs and/or symptoms of meningitis or encephalitis and were therefore included in the analysis. Of the 988 patients admitted with signs and/or symptoms of meningitis and/or encephalitis, 560 (56.7%) were <18 years of age, 330 (33.4%) were 18–64 years of age, and 98 (9.9%) were ≥65 years of age; additionally, 487 (49.3%) were male, 588 (59.5%) were White, and 146 (14.8%) were Black (Table 2). One or more underlying medical condition, most often diabetes (*n* = 51; 5.2%) or cancer (*n* = 33; 3.3%) was present in 123 patients (12.4%) admitted with signs/symptoms of meningitis and/or encephalitis. One or more underlying medical condition was reported by 51 (52.0%) of 98 patients ≥ 65 years of age.

Of the 988 patients admitted with signs/symptoms of meningitis and/or encephalitis, CSF and blood specimens for culture were collected from 751 (76.0%) and 446 (45.1%), respectively; 441 (44.6%) had both CSF and blood specimens collected for culture and 232 (23.5%) had neither CSF nor blood specimens collected for culture (Figure 1). Culture was performed for 712 (94.8%) of the 751 collected CSF samples and 440 (98.7%) of the 446 blood specimens collected for culture, whereas PCR testing for *N. meningitidis* was conducted on 651 (86.7%) of the 751 collected CSF samples. None of the collected blood samples were tested by PCR for *N. meningitidis*. Of the 654 patients hospitalized with meningitis and/or encephalitis in 2017–2023 (when PCR testing of CSF specimens was conducted in the Norton central diagnostic laboratory), 483 (73.8%) had CSF collected and 447 (92.5%) of the 483 CSF specimens were PCR tested for *N. meningitidis*.

Of the 712 patients during the 10-year study period who had CSF culture, 675 (94.8%) received IV antibiotics and 300 (44.4%) of those received antibiotics before CSF collection including 108 (36.0%) who received antibiotics within three hours of CSF collection (Figure 2A). Of the 440 patients who had blood culture, 431 (98.0%) received IV antibiotics and 133 (30.9%) of those received antibiotics before blood collection including 78 (58.6%) within three hours of blood collection and 21 (15.8%) >24 h before blood collection (Figure 2B).

Stratification by age group indicated that compared with younger patients; those ≥65 years of age less frequently received IV antibiotics or had CSF or blood cultured (Table 3). Of the 98 patients in this age group, 57 (58.2%) received IV antibiotics, 54 (55.1%) had CSF cultured, and 26 (26.5%) had blood cultured.

Two (0.2%) of the 988 patients admitted with signs/symptoms of meningitis and/or encephalitis had *N. meningitidis* detected in CSF by culture. The first IMD patient, who had an admission date in 2019, received IV antibiotics 16 min before CSF collection and 9 days before blood collection; the blood sample was negative for *N. meningitidis* by culture. The second IMD patient, who had an admission date in 2020, had only CSF collected, and timing of IV antibiotic receipt was not recorded in the electronic medical record. Admission dates of the two IMD patients were after routine PCR of CSF became available; CSF specimens from both patients were positive for *N. meningitidis* by PCR. Therefore, after routine PCR for *N. meningitidis* became available, 2 (0.4%) of 447 CSF specimens tested by PCR were positive. None of the CSF specimens collected prior to 2017 were tested by PCR for *N. meningitidis*.

## 4. Discussion

This retrospective study of patients admitted with signs and/or symptoms of meningitis and/or encephalitis to five hospitals in Louisville, Kentucky, highlights the potential for under-diagnosis of IMD cases. It should be noted, however, that our study, while identifying the potential for IMD under-diagnosis, did not quantify the extent of IMD under-diagnosis; further studies are needed to achieve that objective. To explore the potential for IMD under-diagnosis, we used hospital admission codes to identify patients admitted to the participating hospitals with signs and/or symptoms of meningitis and/or encephalitis and then confirmed the presence of signs and/or symptoms of meningitis and/or encephalitis at admission by medical record review. Guidelines clearly describe the recommended specimen collection and testing practices for patients with meningitis and/or encephalitis and caution that a specific cause of meningitis and/or encephalitis, be it bacterial, viral or non-infectious, cannot be established without additional testing [9,10,11,12]. We therefore conducted a retrospective review of all patients admitted with signs/symptoms of meningitis/encephalitis, regardless of the postulated etiology at admission, and identified several potential sources for under-diagnosis of IMD cases. First, only 76.0% of patients had CSF collected, whereas guidelines emphasize the importance of CSF collection for diagnosing bacterial meningitis [9,10,11,12]. Additionally, only 45.1% of patients had blood collected, which is also recommended for establishing a specific diagnosis of bacterial meningitis [9,10,11,12]. Second, among patients who had specimens collected and received antibiotics, antibiotics were administered before specimen collection for 44.4% of CSF specimens and 30.9% of blood specimens. Decreased sensitivity of CSF and blood cultures following antibiotic administration is well documented [13,15,16,17,18], and clinical management guidelines stipulate that specimens should be collected before antibiotic administration whenever possible [9,10,11,12]. The decreased sensitivity of cultures in patients receiving antibiotics is an even greater concern in patients who do not have CSF and blood samples tested by PCR.

An additional factor that may have contributed to missed IMD cases is that PCR testing was not performed on CSF specimens until 2017, and even after 2017, PCR testing was performed only on a subset of CSF samples collected. Furthermore, due to the lack of available PCR testing for blood specimens in the United States, none of the blood samples were evaluated using PCR testing. Multiple studies have demonstrated substantial increases in IMD case detection when CSF and blood specimens are tested by PCR in addition to culture, highlighting the importance of testing these specimens by PCR [18,20,22,25]. In England, 42.4% of the 2490 laboratory-confirmed IMD cases reported during April 2017–March 2023 were culture negative but confirmed by PCR [27], Similar results were observed for laboratory-confirmed IMD cases in Italy during 2007–2016; 63.9% of 144 cases confirmed from a CSF sample and 63.6% of 143 cases confirmed from a blood sample were confirmed by PCR only [28]. Additional studies during the early 2000s found that 91.8% of 61 IMD cases in India, 85.0% of 40 IMD cases in Greece, and 92.3% of 13 IMD cases in Fiji were diagnosed by PCR only [29,30,31]. Accordingly, clinical testing guidelines for the management of patients with suspected IMD in several countries recommend PCR testing on blood specimens [9,12,22].

Given the retrospective review nature of this study, results should be interpreted with caution, as medical records may be incomplete. Furthermore, an important limitation of this study is that, although many IMD cases present with signs and/or symptoms of meningitis, a substantial proportion present with septicemia or pneumonia and have no signs or symptoms of meningitis or encephalitis. Our study did not evaluate the potential for under-diagnosis of IMD in patients presenting without meningitis or encephalitis and therefore could not fully assess IMD under-detection. Also, PCR testing of CSF for *N. meningitidis*, did not become available until 2017; even after 2017, CSF was not collected from all patients and not all CSF specimens were tested by PCR indicating a potential for under-diagnosis of IMD. Additionally, although transfer of patients from other hospitals to Norton hospitals is uncommon, procedures performed at healthcare facilities outside of the Norton system would not be included in the EMRs captured by the study; it is therefore possible that some patients had CSF and/or blood specimens collected in other healthcare settings. Another limitation is that in order to evaluate etiological diagnostic practices for patients with signs and/or symptoms of meningitis and/or encephalitis at the participating hospitals, we included patients regardless of symptom severity. These broad inclusion criteria likely resulted in some patients with mild illness (which is less likely to be bacterial), being included which could interfere with our aim to evaluate the potential for under-diagnosis of IMD. Finally, this study was conducted in a single hospital system in Louisville, Kentucky. Further studies would be useful to support the generalization of these findings beyond the participating hospitals.

The aim of this study was to assess the potential for IMD under-diagnosis among patients admitted with signs and/or symptoms of meningitis and/or encephalitis at different hospitals to identify potential under-diagnosis of IMD. Results from this study clearly demonstrate the potential for under-diagnosis of IMD in these participating hospitals; however, the extent of IMD under-diagnosis cannot be estimated. Further studies are needed elsewhere to confirm the potential for, and approximate the extent of, under-diagnosis of IMD cases in the United States.

## 5. Conclusions

This study identified several gaps in specimen collection and laboratory testing among patients admitted with meningitis or encephalitis at five US hospitals during 2014–2023. These gaps may lead to under-diagnosis of IMD, which would result in IMD under-ascertainment by public health surveillance. Further studies are needed to confirm the frequency of IMD under-ascertainment among patients with meningitis or encephalitis, and to explore the potential of IMD under-ascertainment in patients with other IMD presentations, particularly septicemia.

## Figures and Tables

**Figure 1 pathogens-14-00962-f001:**
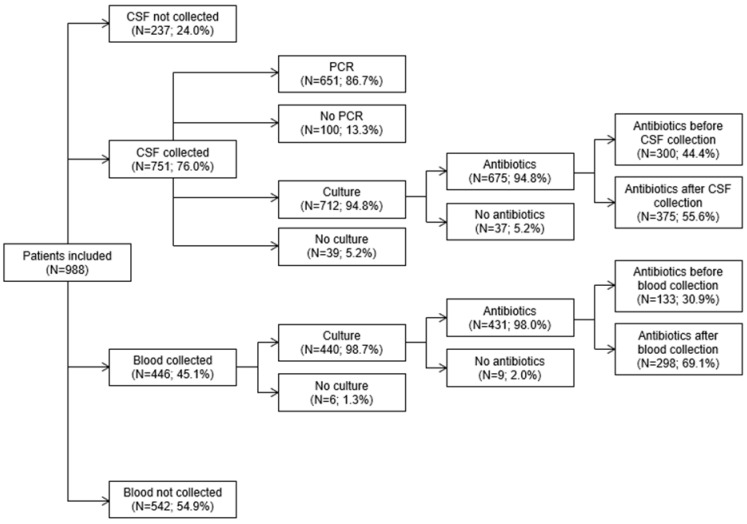
Flow chart of specimen collection, diagnostic tests, and antibiotic receipt. Only patients who had specimens collected and received antibiotics are included in each panel.

**Figure 2 pathogens-14-00962-f002:**
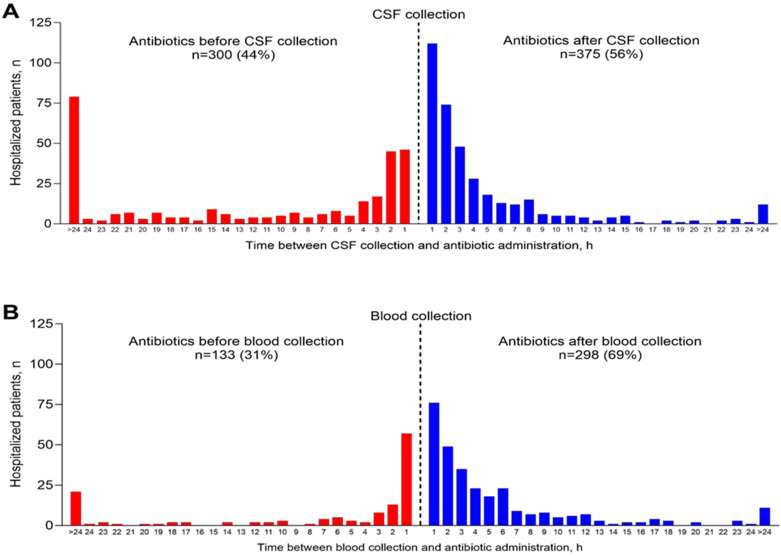
Timing of IV antibiotic administration relative to specimen collection for (**A**) CSF and (**B**).

**Table 1 pathogens-14-00962-t001:** Included ICD-10 DRG codes at admission to one of the five hospitals participating in the study. * Indicates that all codes within that category were included.

Code	Definition
Meningitis	
A17.0	Tuberculous meningitis
A20.3	Plague meningitis
A32.11	Listeria meningitis
A39.0	Meningococcal meningitis
A69.21	Meningitis due to Lyme disease
A87 *	Enteroviral meningitis; adenoviral meningitis; lymphocytic choriomeningitis; other viral meningitis; viral meningitis, unspecified
B01.0	Varicella meningitis
B02.1	Zoster meningitis
B27.02	Gamma herpes viral mononucleosis with meningitis
B27.92	Infectious mononucleosis, unspecified with meningitis
B37.5	Candidal meningitis
B38.4	Coccidioidomycosis meningitis
D86.81	Sarcoid meningitis
G00 *	Hemophilus meningitis; pneumococcal meningitis; streptococcal meningitis; staphylococcal meningitis; other bacterial meningitis; bacterial meningitis, unspecified
G01 *	Meningitis in bacterial diseases classified elsewhere
G02 *	Meningitis in other infectious and parasitic diseases classified elsewhere
G03.0	Nonpyogenic meningitis
G03.8	Meningitis due to other specified causes
G03.9	Meningitis, unspecified
Encephalitis	
A39.81	Meningococcal encephalitis
A83 *	Japanese encephalitis; western equine encephalitis; eastern equine encephalitis; St Louis encephalitis; Australian encephalitis; California encephalitis; Rocio virus disease; other mosquito-borne viral encephalitis; mosquito-borne viral encephalitis, unspecified
A84 *	Far eastern tick-borne encephalitis [Russian spring-summer encephalitis]; central European tick-borne encephalitis; Powassan virus disease; other tick-borne viral encephalitis; tick-borne viral encephalitis, unspecified
A85 *	Enteroviral encephalitis; adenoviral encephalitis; arthropod-borne viral encephalitis, unspecified; other specified viral encephalitis
A86	Unspecified viral encephalitis
G04 *	Acute disseminated encephalitis and encephalomyelitis, unspecified; postinfectious acute disseminated encephalitis and encephalomyelitis (postinfectious ADEM); postimmunization acute disseminated encephalitis, myelitis and encephalomyelitis; tropical spastic paraplegia; bacterial meningoencephalitis and meningomyelitis, not elsewhere classified; acute necrotizing hemorrhagic encephalopathy, unspecified; postinfectious acute necrotizing hemorrhagic encephalopathy; postimmunization acute necrotizing hemorrhagic encephalopathy; other acute necrotizing hemorrhagic encephalopathy; other encephalitis and encephalomyelitis; acute flaccid myelitis; other myelitis; encephalitis and encephalomyelitis, unspecified; myelitis, unspecified
G05 *	Encephalitis and encephalomyelitis in diseases classified elsewhere; myelitis in diseases classified elsewhere

**Table 2 pathogens-14-00962-t002:** Characteristics of patients admitted with signs/symptoms of meningitis and/or encephalitis.

Characteristic, *n* (%)	<18 Years of Age (*n* = 560)	18–64 Years of Age (*n* = 330)	≥65 Years of Age (*n* = 98)	Total (*n* = 988)
Sex				
Male	298 (53.2)	145 (43.9)	44 (44.9)	487 (49.3)
Female	262 (46.8)	185 (56.1)	54 (55.1)	501 (50.7)
Race				
White	353 (63.0)	187 (56.7)	48 (49.0)	588 (59.5)
Black	70 (12.5)	68 (20.6)	8 (8.2)	146 (14.8)
Asian	6 (1.1)	6 (1.8)	3 (3.1)	15 (1.5)
Other	9 (1.6)	5 (1.5)	1 (1.0)	15 (1.5)
Unknown/not reported	122 (21.8)	64 (19.4)	38 (38.8)	224 (22.7)
Ethnicity				
Hispanic/Latino	26 (4.6)	8 (2.4)	1 (1.0)	35 (3.5)
Not Hispanic/Latino	431 (77.0)	260 (78.8)	59 (60.2)	750 (75.9)
Unknown/not reported	103 (18.4)	62 (18.8)	38 (38.8)	203 (20.5)
One or more underlying medical conditions	9 (1.6)	63 (19.1)	51 (52.0)	123 (12.4)
Heart Failure	2 (0.4)	10 (3.0)	14 (14.3)	26 (2.6)
Cerebrovascular disease	3 (0.5)	7 (2.1)	8 (8.2)	18 (1.8)
Renal disease	1 (0.2)	9 (2.7)	11 (11.2)	21 (2.1)
Liver disease	0	7 (2.1)	0	7 (0.7)
Diabetes	3 (0.5)	29 (8.8)	19 (19.4)	51 (5.2)
Asplenia	0	1 (0.3)	0	1 (0.1)
Chronic obstructive pulmonary disease	0	10 (3.0)	12 (12.2)	22 (2.2)
HIV infection	0	7 (2.1)	1 (1.0)	8 (0.8)
Cancer	2 (0.4)	14 (4.2)	17 (17.3)	33 (3.3)
Two or more underlying medical conditions	1 (0.2)	23 (7.0)	24 (24.5)	48 (4.9)

**Table 3 pathogens-14-00962-t003:** Specimen collection, diagnostic tests, and antibiotic receipt overall and by age group.

Characteristic, *n* (%)	<18 Years (*n* = 560)	18–64 Years (*n* = 330)	≥65 Years (*n* = 98)	Total (*n* = 988)
Antibiotics	428 (76.4)	243 (73.6)	57 (58.2)	728 (73.7)
CSF collected	440 (78.6)	254 (77.0)	57 (58.2)	751 (76.0)
CSF PCR for *N. meningitidis*	415 (74.1)	186 (56.4)	50 (51.0)	651 (65.9)
CSF cultured	415 (74.1)	243 (73.6)	54 (55.1)	712 (72.1)
Antibiotics before	162 (28.9)	99 (30.0)	39 (39.8)	300 (30.4)
Antibiotics after	237 (42.3)	125 (37.9)	13 (13.3)	375 (38.0)
Blood collected	307 (54.8)	113 (34.2)	26 (26.5)	446 (45.1)
Blood cultured	301 (53.8)	113 (34.2)	26 (26.5)	440 (44.5)
Antibiotics before	93 (16.6)	29 (8.8)	11 (11.2)	133 (13.5)
Antibiotics after	204 (36.4)	79 (23.9)	15 (15.3)	298 (30.2)
Both CSF and blood collected	306 (54.6)	109 (33.0)	26 (26.5)	441 (44.6)
Neither CSF nor blood collected	119 (21.3)	72 (21.8)	41 (41.8)	232 (23.5)

## Data Availability

Upon request, and subject to review, Pfizer will provide summary data. See https://www.pfizer.com/science/clinical-trials/trial-data-and-results (accessed on 2 August 2025) for more information.

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
