# Peer review of "A Ten-Year Retrospective Review of Medical Records of Patients Admitted with Meningitis or Encephalitis at Five Hospitals in the United States Highlights the Potential for Under-Ascertainment of Invasive Meningococcal Disease"

_pathogens, 2025, doi:10.3390/pathogens14100962_

Round 1

Reviewer 1 Report

Comments and Suggestions for Authors

The authors included in the study only patients with ICD-10 codes for meningitis or encephalitis, excluding IMD cases presenting with septicemia without CNS involvement. While the authors mention this in the limitations section, the consequences of this choice—namely, the potential substantial underestimation of IMD incidence—should be more strongly emphasized and supported by literature. The rationale for excluding septic cases is not entirely clear from a reviewer’s perspective. This is inconsistent with the paper’s title, which refers to “invasive meningococcal disease,” a condition that by definition is septic. The authors should consider changing the title to indicate that the paper focuses on meningitis.

2. PCR became routinely available at the study center in 2017; however, there is no detailed comparison of detection rates before and after this date. Such an analysis could substantially enhance the paper’s value by demonstrating the impact of molecular diagnostics on pathogen detection rates. Furthermore, the study does not assess whether earlier specimen collection or the use of PCR translated into improved pathogen detection or better patient outcomes. These data would be valuable for drawing practical conclusions.

3. The data originate from a single hospital system in Louisville, which limits the generalizability of the findings to the national population. This limitation should be clearly highlighted in the discussion, as it does not fully justify generalizing the conclusions.

4. The sponsor’s involvement in study design, data analysis, and manuscript preparation is substantial. In accordance with good publication practice, such information should be presented earlier (e.g., in the introduction or methods) to ensure that readers are aware of this context before interpreting the results.

5. Although the authors acknowledge the possibility of underdiagnosis of IMD, they do not attempt to estimate the scale of this phenomenon. It would be advisable to rely on data from the literature or other epidemiological surveillance systems to approximate the potential magnitude of the problem.

6. There are numerous double spaces and awkward line breaks, particularly noticeable in the title, author list, and ICD code tables.

7. Some table headers are split across lines (e.g., “Men-” / “ingitis”, “En-” / “cephalitis”).

8. Abbreviations are not introduced consistently. In accordance with good practice, every abbreviation should be defined at first mention, including in the abstract.

9. The abstract is overloaded with numerical data, making it difficult to read fluently. It could be simplified by focusing on key findings.

10. There is content repetition between the abstract, introduction, and discussion (e.g., information about decreased culture sensitivity after antibiotic use).

Reviewer’s suggestions:

  • Attempt to estimate the number of undetected IMD cases based on available sources, or reconsider the study’s underlying assumption. The term “invasive meningococcal disease” corresponds to a different ICD classification.

  • Standardize the use of abbreviations and expand each at first mention, including in the abstract.

  • Improve table and text formatting by removing unnecessary line breaks.

  • Reduce redundancy and streamline the abstract.

  • Include information on the sponsor’s role at the very beginning of the manuscript.

  • If possible, expand the analysis to include IMD cases without meningitis or encephalitis.

Author Response

Responses to reviewers’ comments

Title: A ten-year retrospective review of medical records of patients admitted with meningitis or encephalitis at five hospitals in the United States highlights the potential for under-ascertainment of invasive meningococcal disease

Manuscript ID: pathogens-3826819

Line numbers mentioned in the response to reviewers’ comments relate to the line numbers in the CLEAN version of the manuscript. Text that is from the revised manuscript are written in italics and in quotation marks.

Reviewer #1

  1. The authors included in the study only patients with ICD-10 codes for meningitis or encephalitis, excluding IMD cases presenting with septicemia without CNS involvement. While the authors mention this in the limitations section, the consequences of this choice—namely, the potential substantial underestimation of IMD incidence—should be more strongly emphasized and supported by literature. The rationale for excluding septic cases is not entirely clear from a reviewer’s perspective. This is inconsistent with the paper’s title, which refers to “invasive meningococcal disease,” a condition that by definition is septic. The authors should consider changing the title to indicate that the paper focuses on meningitis.

Response 1: Thank you for your helpful comment. We would like to address several items raised in your comment. First, this study is not a retrospective review of patients with IMD; this is a retrospective review of patients who present with signs and/or symptoms of meningitis and/or encephalitis upon admission at the participating hospitals. Therefore, it is incorrect to say that we excluded “IMD cases presenting with septicemia without CNS involvement.” Our only exclusion was of patients admitted without signs and/or symptoms of meningitis recorded in their medical records. This is explained in the methods section at lines 95-100:

“Patients were included if they were admitted to the hospital with one or more of the admission codes, using the International Classification of Diseases, Tenth Revision (ICD-10) Diagnosis-Related Group (DRG) codes, for meningitis or encephalitis (Table 1). Corresponding clinical signs and symptoms (ie, fever, headache, stiff neck, sensitivity to light, nausea, vomiting, confusion, etc) recorded in the EMRs of these patients were reviewed by study physicians; patients without such signs and/or symptoms were excluded from the analysis.”

Second, your comment indicates a need to more clearly explain the purpose of our study. The aim of our study was to determine if there is a potential for under-detection of IMD among patients admitted with signs and/or symptoms of meningitis and/or encephalitis. We, therefore, revised the stated aim of the study in the introduction section at lines 83-86:

“This study aimed to analyze IMD laboratory confirmation methods for patients presenting with signs and/or symptoms of meningitis and/or encephalitis and admitted to one of five hospitals in Louisville, Kentucky, and assess the potential for IMD under-diagnosis and potential implications for meningococcal disease surveillance.”

Furthermore, to emphasize the aim of the study, we added text in the discussion section at lines 172-179:

“We used hospital admission codes to identify patients admitted to the participating hospitals with signs and/or symptoms of meningitis and/or encephalitis and then confirmed the presence of signs and/or symptoms of meningitis and/or encephalitis at admission by medical record review. Guidelines clearly describe the recommended specimen collection and testing practices for patients with meningitis and/or encephalitis and caution that a specific cause of meningitis and/or encephalitis, be it bacterial, viral or non-infectious, cannot be established without additional testing [9-12]. We therefore conducted a retrospective review of all patients admitted with signs/symptoms of meningitis/encephalitis, regardless of the postulated etiology at admission, and identified several potential sources for under-diagnosis of IMD cases.”

Additionally, we revised the discussion section to emphasize that purpose of the study and to emphasize that estimating the extent of under-ascertainment of IMD cases was beyond the purpose of our study. This revision of the discussion section is at lines 220-225:

 “The aim of this study was to assess the potential for IMD under-diagnosis among patients admitted with signs and/or symptoms of meningitis and/or encephalitis at different hospitals to identify potential under-diagnosis of IMD. Results from this study clearly demonstrate the potential for under-diagnosis of IMD in these participating hospitals, however, the extent of IMD under-diagnosis cannot be estimated. Further studies are needed elsewhere to confirm the potential for, and approximate the extent of, under-diagnosis of IMD cases in the United States.”

Finally, thank you also for suggesting that we revise the title of the article; we agree that revising the title of the article may reduce the likelihood of readers making similar misunderstandings about the population of patients included in our study. To be specific about the population included in our study, our revised title is:

 “A ten-year retrospective review of medical records of patients admitted with meningitis or encephalitis at five hospitals in the United States highlights the potential for under-ascertainment of invasive meningococcal disease”

  1. PCR became routinely available at the study center in 2017; however, there is no detailed comparison of detection rates before and after this date. Such an analysis could substantially enhance the paper’s value by demonstrating the impact of molecular diagnostics on pathogen detection rates. Furthermore, the study does not assess whether earlier specimen collection or the use of PCR translated into improved pathogen detection or better patient outcomes. These data would be valuable for drawing practical conclusions.

Response 2: Thank you for the suggestion to add additional information about the two IMD cases diagnosed during the study period. We revised the results section at lines 165-168:

“Admission dates of the two IMD patients were after routine PCR of CSF became available; CSF specimens from both patients were positive for N. meningitidis by PCR. Therefore, after routine PCR for N meningitidis became available, 2 (0.4%) of 447 CSF specimens tested by PCR were positive. None of the CSF specimens collected prior to 2017 were tested by PCR for N meningitidis.”

  1. The data originate from a single hospital system in Louisville, which limits the generalizability of the findings to the national population. This limitation should be clearly highlighted in the discussion, as it does not fully justify generalizing the conclusions.

Response 3: Our study was conducted in five hospitals in Louisville, Kentucky. The setting of our study is introduced in the introduction section and then well described in the methods section at lines 89-94

“This study was a retrospective review of the electronic medical records (EMRs) of patients with signs and/or symptoms of meningitis and/or encephalitis who were admitted to five Norton hospitals in Louisville, Kentucky from 2014 to 2023. The five participating hospitals have a combined total of 2,204 beds and had almost 100,000 hospital admissions per year during the study period. The hospitals included Norton Audubon Hospital (432 beds), Norton Brownsboro Hospital (432), Norton Children’s Hospital (372), Norton Hospital (605), and Norton Women’s and Children’s Hospital (363).”

We agree that caution should be taken in generalizing the results of our study to other hospitals in the United States. We therefore have added text to our statement about generalization in the discussion section at lines 217-219:

“Finally, this study was conducted in a single hospital system in Louisville, Kentucky. Further studies would be useful to support the generalization of these findings beyond the participating hospitals.” 

  1. The sponsor’s involvement in study design, data analysis, and manuscript preparation is substantial. In accordance with good publication practice, such information should be presented earlier (e.g., in the introduction or methods) to ensure that readers are aware of this context before interpreting the results.

Response 4: Several of the authors are employees of Pfizer; this is stated in the author affiliations listed on lines 8-19. We followed the journal directions, in the instructions to submitting authors, and placed the sponsor involvement in the appropriate section, entitled “Funding” at line 239:

Funding: This work and publication costs were supported by Pfizer Inc.”

Furthermore, as requested, the involvement of Pfizer in the study design, data analysis, and manuscript preparation are stated, as instructed in the journal directions to submitting authors, at lines 248-250:

Conflicts of Interest: JR, SF, TC, and JP received research funding from Pfizer. All other authors are current or former employees of Pfizer and may hold stock or stock options. Pfizer was involved in choosing the research project; study design; collection, analyses, and interpretation of data; writing the manuscript; and the decision to publish the results.”

  1. Although the authors acknowledge the possibility of underdiagnosis of IMD, they do not attempt to estimate the scale of this phenomenon. It would be advisable to rely on data from the literature or other epidemiological surveillance systems to approximate the potential magnitude of the problem.

Response 5: As mentioned in our response to comment #1, the goal of this study was to identify the potential for under-diagnosis of IMD among patients admitted with signs and/or symptoms of meningitis and/or encephalitis. The study clearly identifies that potential for under-diagnosis of IMD in the participating hospitals during the study period. However, estimating the extent of the under-diagnosis of IMD is beyond the scope of this study. To ensure this is clear to the reader, we revised the discussion section at lines 220-225:

“The aim of this study was to assess the potential for IMD under-diagnosis among patients admitted with signs and/or symptoms of meningitis and/or encephalitis at different hospitals to identify potential under-diagnosis of IMD. Results from this study clearly demonstrate the potential for under-diagnosis of IMD in these participating hospitals, however, the extent of IMD under-diagnosis cannot be estimated. Further studies are needed elsewhere to confirm the potential for, and approximate the extent of, un-der-diagnosis of IMD cases in the United States.”

  1. There are numerous double spaces and awkward line breaks, particularly noticeable in the title, author list, and ICD code tables.

Response 6: Thank you for your comment; we have revised the manuscript accordingly.

  1. Some table headers are split across lines (e.g., “Men-” / “ingitis”, “En-” / “cephalitis”).

Response 7: As requested, we have revised the headers in the tables.

  1. Abbreviations are not introduced consistently. In accordance with good practice, every abbreviation should be defined at first mention, including in the abstract.

Response 8: As requested, we have ensured that all abbreviations are defined only once and are listed in the abbreviation section at lines 251-261.

  1. The abstract is overloaded with numerical data, making it difficult to read fluently. It could be simplified by focusing on key findings.

Response 9: As suggested, we simplified the abstract at lines 22-36:

“Laboratory confirmation of invasive meningococcal disease (IMD) relies on detection of Neisseria meningitidis in a biological specimen. Clinical management guidelines for patients presenting with signs and/or symptoms of meningitis and encephalitis emphasize the need for appropriate specimen collection for laboratory testing. To explore the potential for IMD under-diagnosis, we reviewed medical records of patients admitted with signs and/or symptoms of meningitis or encephalitis at five hospitals in Louisville, Kentucky, in 2014 to 2023. Among 675 patients admitted with meningitis and/or encephalitis with cerebrospinal fluid (CSF) cultured who received antibiotics, 300 (44.4%) received antibiotics before CSF collection. Among 431 with blood cultured who received antibiotics, 133 (30.9%) received antibiotics before blood collection. Among 751 patients with CSF collected, 651 (86.7%) CSF specimens were tested using polymerase chain reaction (PCR) for N meningitidis detection. No blood specimens were PCR-tested. These findings indicated that current standard-of-care practices may lead to IMD under-diagnosis. Since public health surveillance relies on IMD laboratory diagnosis, these findings highlight the potential for under-ascertained IMD by surveillance.”

  1. There is content repetition between the abstract, introduction, and discussion (e.g., information about decreased culture sensitivity after antibiotic use).

Response 10: As requested, we condensed the abstract. We also reduced the content repetition between the introduction and discussion.

  1. Attempt to estimate the number of undetected IMD cases based on available sources, or reconsider the study’s underlying assumption. The term “invasive meningococcal disease” corresponds to a different ICD classification.

Response 11: As explained in our response to comment #5, we added the following text in the discussion section at lines 220-225:

“The aim of this study was to assess the potential for IMD under-diagnosis among patients admitted with signs and/or symptoms of meningitis and/or encephalitis at different hospitals to identify potential under-diagnosis of IMD. Results from this study clearly demonstrate the potential for under-diagnosis of IMD in these participating hospitals, however, the extent of IMD under-diagnosis cannot be estimated. Further studies are needed elsewhere to confirm the potential for, and approximate the extent of, un-der-diagnosis of IMD cases in the United States.”

  1. Standardize the use of abbreviations and expand each at first mention, including in the abstract.

Response 12: Please see response to comment #8.

  1. Improve table and text formatting by removing unnecessary line breaks.

Response 13: Please see response to comment #7.

  1. Reduce redundancy and streamline the abstract.

Response 14: Please see response to comment #9.

  1. Include information on the sponsor’s role at the very beginning of the manuscript.

Response 15: Please see response to comment #4.

  1. If possible, expand the analysis to include IMD cases without meningitis or encephalitis

Response 16: As explained in our response to comment #1, this study is not a retrospective review of patients with IMD; this is a retrospective review of patients who present with signs and/or symptoms of meningitis and/or encephalitis upon admission at the participating hospitals. Expanding the analysis to include patients admitted to the participating hospitals with signs and/or symptoms other than meningitis and/or encephalitis is beyond the scope of the study.

Reviewer 2 Report

Comments and Suggestions for Authors

The authors presented an important project auditing management of patients with suspected meningitis or encephalitis.

Unfortunately the authors included diagnostic codes of viral meningitis. A mild viral meningitis will be less likely to trigger a rigorous guideline based management with early lumbar puncture and doing blood cultures and PCR for Neisseria meningitis on peripheral blood. The authors have not explained the break down of the final diagnoses in their patients and what was the break down of clinical features including petechial rash, septic shock, CRP above 100 or elevated lactate levels.

There were only 2 patients with confirmed meningococcal disease. This means the study cannot draw any conclusions regarding ascertainment of meningococcal disease

Author Response

Responses to reviewers’ comments

Title: A ten-year retrospective review of medical records of patients admitted with meningitis or encephalitis at five hospitals in the United States highlights the potential for under-ascertainment of invasive meningococcal disease

Manuscript ID: pathogens-3826819

Line numbers mentioned in the response to reviewers’ comments relate to the line numbers in the CLEAN version of the manuscript. Text that is from the revised manuscript are written in italics and in quotation marks.

 Reviewer #2

  1. The authors presented an important project auditing management of patients with suspected meningitis or encephalitis.

Response 1: Thank you for your kind comments and for your helpful review. To be precise, please note that our study is not of “patients with suspected meningitis or encephalitis.” Our study is of patients admitted to a participating hospital with signs and/or symptoms of meningitis and/or encephalitis. To make this clear, we have revised the title of the article to:

“A ten-year retrospective review of medical records of patients admitted with meningitis or encephalitis at five hospitals in the United States highlights the potential for under-ascertainment of invasive meningococcal disease”

Furthermore, we emphasized the study population in our discussion section at lines 170-179:

“This retrospective study of patients admitted with signs and/or symptoms of meningitis and/or encephalitis to five hospitals in Louisville, Kentucky, highlights the potential for under-diagnosis of IMD cases. We used hospital admission codes to identify patients admitted to the participating hospitals with signs and/or symptoms of meningitis and/or encephalitis and then confirmed the presence of signs and/or symptoms of meningitis and/or encephalitis at admission by medical record review. Guidelines clearly describe the recommended specimen collection and testing practices for patients with meningitis and/or encephalitis and caution that a specific cause of meningitis and/or encephalitis, be it bacterial, viral or non-infectious, cannot be established without additional testing [9-12]. We therefore conducted a retrospective review of all patients admitted with signs/symptoms of meningitis/encephalitis, regardless of the postulated etiology at admission, and identified several potential sources for under-diagnosis of IMD cases.”

  1. Unfortunately the authors included diagnostic codes of viral meningitis. A mild viral meningitis will be less likely to trigger a rigorous guideline based management with early lumbar puncture and doing blood cultures and PCR for Neisseria meningitis on peripheral blood.

Response 2: Thank you for your comment; we agree. Therefore, we have added the following statement in the limitations paragraph in the discussion section at lines 212-217:

“Another limitation is that in order to evaluate etiological diagnostic practices for patients with signs and/or symptoms of meningitis and/or encephalitis at the participating hospitals, we included patients regardless of symptom severity. These broad inclusion criteria likely resulted in some patients with mild illness (which is less likely to be bacterial), being included which could interfere with our aim to evaluate the potential for under-diagnosis of IMD.”

  1. The authors have not explained the break down of the final diagnoses in their patients and what was the break down of clinical features including petechial rash, septic shock, CRP above 100 or elevated lactate levels.

Response 3: Thank you for the comment but we submit that the final diagnosis of the patients in the study population is not relevant. The study population is all patients admitted to participating hospitals during the study period with signs and/or symptoms of meningitis and/or encephalitis. As emphasized in the meningitis and encephalitis management guidelines, diagnosing the specific etiology of the meningitis and/or encephalitis is not possible without additional testing. Therefore, the management guidelines are for all etiologies that result in a patient presenting with signs and/or symptoms of meningitis and/or encephalitis.

  1. There were only 2 patients with confirmed meningococcal disease. This means the study cannot draw any conclusions regarding ascertainment of meningococcal disease.

Response 4: The aim of this study is not to measure the extent of the under-diagnosis of IMD. As explained in the introduction section at lines 83-86:

“This study aimed to analyze IMD laboratory confirmation methods for patients presenting with signs and/or symptoms of meningitis and/or encephalitis and admitted to one of five hospitals in Louisville, Kentucky, and assess the potential for IMD under-diagnosis and potential implications for meningococcal disease surveillance.”

To make it clear that the aim of the study is not to draw conclusions about the extent of the under-diagnosis of IMD, we have revised the discussion section at lines 220-225:

 “The aim of this study was to assess the potential for IMD under-diagnosis among patients admitted with signs and/or symptoms of meningitis and/or encephalitis at different hospitals to identify potential under-diagnosis of IMD. Results from this study clearly demonstrate the potential for under-diagnosis of IMD in these participating hospitals, however, the extent of IMD under-diagnosis cannot be estimated. Further studies are needed elsewhere to confirm the potential for, and approximate the extent of, under-diagnosis of IMD cases in the United States.”

“Patients were included if they were admitted to the hospital with one or more of the admission codes, using the International Classification of Diseases, Tenth Revision (ICD-10) Diagnosis-Related Group (DRG) codes, for meningitis or encephalitis (Table 1). Corresponding clinical signs and symptoms (ie, fever, headache, stiff neck, sensitivity to light, nausea, vomiting, confusion, etc) recorded in the EMRs of these patients were reviewed by study physicians; patients without such signs and/or symptoms were excluded from the analysis.”

Reviewer 3 Report

Comments and Suggestions for Authors

General Comments

This study retrospectively reviewed patients admitted to five hospitals with a diagnosis of meningitis/encephalitis. The authors categorized patients as "treated" or "untreated" before antibiotics were administered, and whether or not CSF or blood cultures were collected. They found that because PCR was performed in a limited number of cases, the incidence of invasive meningococcal disease (IMD) may have been underestimated. The overall manuscript is well-written; I just have a few questions and comments.

Specific Comments

  1. N. meningitidis should be italicized throughout the manuscript. When it first appears, it should be spelled out in full in the abstract.
  2. L107-109: I encourage the authors to provide a summary table of clinical signs and symptoms.
  3. Table 1: This table raises a concern that some ICD codes specifically refer to a certain pathogen, such as Japanese encephalitis. Including patients with a definite diagnosis seems to imply that these patients might have been coinfected with N. meningitidis. I believe this hypothesis could overestimate the incidence of IMD. I suggest providing the number of patients for each ICD code in this table. Excluding patients with a definite diagnosis could also be an alternative method. If medical charts were reviewed, a more detailed analysis could potentially be performed.
  4. I am also curious about the aim of this manuscript. Why focus on IMD only? Why do not choose another pathogen, or include more than one pathogen? Other pathogens, such as Group B Streptococcus, may also be sensitive to antibiotic treatment. Providing more information on the knowledge gaps and rationale for this study would make the results more convincing.

Author Response

Responses to reviewers’ comments

Title: A ten-year retrospective review of medical records of patients admitted with meningitis or encephalitis at five hospitals in the United States highlights the potential for under-ascertainment of invasive meningococcal disease

Manuscript ID: pathogens-3826819

Line numbers mentioned in the response to reviewers’ comments relate to the line numbers in the CLEAN version of the manuscript. Text that is from the revised manuscript are written in italics and in quotation marks.

 Reviewer #3

GENERAL COMMENT: This study retrospectively reviewed patients admitted to five hospitals with a diagnosis of meningitis/encephalitis. The authors categorized patients as "treated" or "untreated" before antibiotics were administered, and whether or not CSF or blood cultures were collected. They found that because PCR was performed in a limited number of cases, the incidence of invasive meningococcal disease (IMD) may have been underestimated. The overall manuscript is well-written; I just have a few questions and comments.

Response: Thank you for your kind comments and for your helpful review. An important clarification is that this study is not a study of patients diagnosed with meningitis or encephalitis; this study is of patients admitted to the participating hospitals with signs and/or symptoms of meningitis and/or encephalitis. To make this clear, we have revised the title of the article to:

“A ten-year retrospective review of medical records of patients admitted with meningitis or encephalitis at five hospitals in the United States highlights the potential for under-ascertainment of invasive meningococcal disease”

SPECIFIC COMMENTS

1. meningitidis should be italicized throughout the manuscript. When it first appears, it should be spelled out in full in the abstract.

Response 1: Thank you for identifying this error. We have made this suggested correction throughout the manuscript.

2. L107-109: I encourage the authors to provide a summary table of clinical signs and symptoms.

Response 2: Thank you for your suggestion. All patients in the study population had signs and/or symptoms of meningitis and/or encephalitis upon admission to the participating hospitals during the study period. This is clarified in the methods section at lines 95-100:

“Patients were included if they were admitted to the hospital with one or more of the admission codes, using the International Classification of Diseases, Tenth Revision (ICD-10) Diagnosis-Related Group (DRG) codes, for meningitis or encephalitis (Table 1). Corresponding clinical signs and symptoms (ie, fever, headache, stiff neck, sensitivity to light, nausea, vomiting, confusion, etc) recorded in the EMRs of these patients were reviewed by study physicians; patients without such signs and/or symptoms were excluded from the analysis.”

Since the purpose of the study is to understand if the specimen collection and testing of the study population (i.e., patients admitted with signs and/or symptoms of meningitis and/or encephalitis), we submit that the specific signs and symptoms of the patients in the study population are not relevant provided that all patients in the study population were admitted with signs and/or symptoms of meningitis and/or encephalitis. 

3. Table 1: This table raises a concern that some ICD codes specifically refer to a certain pathogen, such as Japanese encephalitis. Including patients with a definite diagnosis seems to imply that these patients might have been coinfected with N. meningitidis. I believe this hypothesis could overestimate the incidence of IMD. I suggest providing the number of patients for each ICD code in this table. Excluding patients with a definite diagnosis could also be an alternative method. If medical charts were reviewed, a more detailed analysis could potentially be performed.

Response 3: Thank you for your comment. However, your comment indicates a misunderstanding about the study population of our study. This study is not a retrospective review of patients with IMD; this is a retrospective review of patients who present with signs and/or symptoms of meningitis and/or encephalitis upon admission to the participating hospitals. We, therefore, used the admitting diagnosis codes to help identify the patients who were admitted with suspected meningitis and/or encephalitis. We then reviewed the medical records of these patients included only patients with signs and/or symptoms of meningitis and/or encephalitis. This is described in the methods section at lines 95-100:

“Patients were included if they were admitted to the hospital with one or more of the admission codes, using the International Classification of Diseases, Tenth Revision (ICD-10) Diagnosis-Related Group (DRG) codes, for meningitis or encephalitis (Table 1). Corresponding clinical signs and symptoms (ie, fever, headache, stiff neck, sensitivity to light, nausea, vomiting, confusion, etc) recorded in the EMRs of these patients were reviewed by study physicians; patients without such signs and/or symptoms were excluded from the analysis.”

To further clarify our study population, we revised the title to:

 “A ten-year retrospective review of medical records of patients admitted with meningitis or encephalitis at five hospitals in the United States highlights the potential for under-ascertainment of invasive meningococcal disease”

4. I am also curious about the aim of this manuscript. Why focus on IMD only? Why do not choose another pathogen, or include more than one pathogen? Other pathogens, such as Group B Streptococcus, may also be sensitive to antibiotic treatment. Providing more information on the knowledge gaps and rationale for this study would make the results more convincing.

Response 4: We have clarified the aim of the study by revising the text in the discussion section at lines 83-86:

“This study aimed to analyze IMD laboratory confirmation methods for patients presenting with signs and/or symptoms of meningitis and/or encephalitis and admitted to one of five hospitals in Louisville, Kentucky, and assess the potential for IMD under-diagnosis and potential implications for meningococcal disease surveillance.”

The study population of our study was not patients with IMD; your question of “why focus on IMD” is based on an incorrect understanding. The focus of our study was not patients with IMD. Our study population, and therefore the focus of our study was on, patients admitted with signs and/or symptoms of meningitis and/or encephalitis. We introduce this in the introduction section and more precisely in the methods section at lines 95-100:

“Patients were included if they were admitted to the hospital with one or more of the admission codes, using the International Classification of Diseases, Tenth Revision (ICD-10) Diagnosis-Related Group (DRG) codes, for meningitis or encephalitis (Table 1). Corresponding clinical signs and symptoms (ie, fever, headache, stiff neck, sensitivity to light, nausea, vomiting, confusion, etc) recorded in the EMRs of these patients were reviewed by study physicians; patients without such signs and/or symptoms were excluded from the analysis.”

With the goal of identifying the potential for under-diagnosis of IMD among patients admitted with signs and/or symptoms of meningitis and/or encephalitis, the study population for this study was all patients admitted to the participating hospitals with signs and/or symptoms of meningitis and/or symptoms of meningitis and/or encephalitis. To identify patients admitted with signs and/or symptoms of meningitis and/or encephalitis, we first screened the hospital records for all patients admitted with any admitting diagnosis that could manifest with a sign and/or symptom of meningitis and/or encephalitis. For this reason, we included suspected viral causes of meningitis and/or encephalitis among the suspected admitting diagnosis (to identify all patients admitted with a sign and/or symptom of meningitis and/or encephalitis). To ensure this is clear to the reader, we revised the text in the discussion section at lines 172-179:

 “We used hospital admission codes to identify patients admitted to the participating hospitals with signs and/or symptoms of meningitis and/or encephalitis and then confirmed the presence of signs and/or symptoms of meningitis and/or encephalitis at admission by medical record review. Guidelines clearly describe the recommended specimen collection and testing practices for patients with meningitis and/or encephalitis and caution that a specific cause of meningitis and/or encephalitis, be it bacterial, viral or non-infectious, cannot be established without additional testing [9-12]. We therefore conducted a retrospective review of all patients admitted with signs/symptoms of meningitis/encephalitis, regardless of the postulated etiology at admission, and identified several potential sources for under-diagnosis of IMD cases.”

Round 2

Reviewer 1 Report

Comments and Suggestions for Authors

The authors made the suggested changes.

Author Response

Thank you for your review of our manuscript. Your comments have improved the manuscript. 

Reviewer 2 Report

Comments and Suggestions for Authors

The authors were unable to improve their investigation sufficiently.

Author Response

Thank you for your review of our manuscript. Your comments and suggestions have improved the manuscript.

Reviewer comment

  1. The authors declared that they wanted to investigate the diagnosis of meningococcal diseases. The inclusion criteria were inappropriately containing patients with mild viral meningitis. This is a flawed approach leading to erroneous assumptions regarding underascertainment. The fact that the authors did not use a recognized clinical case definition of IMD to assess ascertainment of this diagnosis renders this study invalid.

Response: We agree that that our study would have used a “flawed approach” and that the results of our study would be “invalid” if the objective our study was to “investigate the diagnosis of meningococcal diseases.” However, the objective of our manuscript was NOT to “investigate the diagnosis of meningococcal disease”. The objective of our manuscript was to investigate patients hospitalized with signs and/or symptoms of meningitis and/or encephalitis to explore the potential for under-ascertainment of invasive meningococcal disease. The objective of our manuscript is stated in the title at lines 2-5:

“A ten-year retrospective review of medical records of patients admitted with meningitis or encephalitis at five hospitals in the United States highlights the potential for under-ascertainment of invasive meningococcal disease”

The objective of our manuscript is stated in the abstract at lines 25-28:

“To explore the potential for IMD under-diagnosis, we reviewed medical records of patients admitted with signs and/or symptoms of meningitis or encephalitis at five hospitals in Louisville, Kentucky, in 2014 to 2023.”

The objective of our manuscript is also stated in the introduction section at lines 83-86:

“This study aimed to analyze IMD laboratory confirmation methods for patients presenting with signs and/or symptoms of meningitis and/or encephalitis and admitted to one of five hospitals in Louisville, Kentucky, and assess the potential for IMD under-diagnosis and potential implications for meningococcal disease surveillance.”

The objective is reiterated in the discussion section at lines 170-171 :

“This retrospective study of patients admitted with signs and/or symptoms of meningitis and/or encephalitis to five hospitals in Louisville, Kentucky, highlights the potential for under-diagnosis of IMD cases.”

The objective is also restated in the discussion section at lines 222-224:

“The aim of this study was to assess the potential for IMD under-diagnosis among patients admitted with signs and/or symptoms of meningitis and/or encephalitis at different hospitals to identify potential under-diagnosis of IMD.”

To ensure that readers understand that the objective is NOT to investigate the diagnosis of meningococcal diseases (but to investigate the potential for under-diagnosis of IMD), we added text to the first paragraph of the discussion section at lines 172-174:

“It should be noted, however, that our study, while identifying the potential for IMD under-diagnosis, did not quantify the extent of IMD under-diagnosis; further studies are needed to achieve that objective. To explore the potential for IMD under-diagnosis, we ……………….”

Reviewer 3 Report

Comments and Suggestions for Authors

The authors addressed the issues I mentioned. I have no further comments.

Author Response

Thank you for your review of our manuscript; your comments and suggestions have improved the manuscript.